# Mass Spectrometry-Based Proteomics: Analyses Related to Drug-Resistance and Disease Biomarkers

**DOI:** 10.3390/medicina59101722

**Published:** 2023-09-27

**Authors:** Marco Agostini, Pietro Traldi, Mahmoud Hamdan

**Affiliations:** Istituto di Ricerca Pediatrica Città della Speranza, Corso Stati Uniti 4, 35100 Padova, Italy; m.agostini@unipd.it (M.A.); mhglaxo@gmail.com (M.H.)

**Keywords:** mass spectrometry, post-translational modifications, drug resistance

## Abstract

Mass spectrometry-based proteomics is a key player in research efforts to characterize aberrant epigenetic alterations, including histone post-translational modifications and DNA methylation. Data generated by this approach complements and enrich datasets generated by genomic, epigenetic and transcriptomics approaches. These combined datasets can provide much-needed information on various mechanisms responsible for drug resistance, the discovery and validation of potential biomarkers for different diseases, the identification of signaling pathways, and genes and enzymes to be targeted by future therapies. The increasing use of high-resolution, high-accuracy mass spectrometers combined with more refined protein labeling and enrichment procedures enhanced the role of this approach in the investigation of these epigenetic modifications. In this review, we discuss recent MS-based studies, which are contributing to current research efforts to understand certain mechanisms behind drug resistance to therapy. We also discuss how these MS-based analyses are contributing to biomarkers discovery and validation.

## 1. Introduction

Almost every cell in the human body contains the chromatin macromolecular structure. The main component in such a structure is the nucleosome, which consists of a central histone octamer of four core proteins (H3, H4, H2A and H2B) wrapped around 147 base pairs of DNA [1]. For a number of years, chromatin was considered an inert packaging structure, but more recent studies consider it a dynamic scaffold capable of responding to specific signals and regulating the accessibility of the DNA to the various components of the cellular machinery [2]. Histone proteins have relatively low molecular weights ranging from 10 to 15 kDa, which facilitates their MS investigation using different ionization techniques and accurate mass measurements to investigate the intact proteins.

It is well recognized that amino acids in protein side chains can experience different modifications following their biosynthesis. So far there are more than 400 known post-translational modifications (PTMs), some of which are known to affect the function(s) of these modified proteins [3]. Histone proteins are known to experience a number of covalent PTMs (often referred to as epigenetic marks), these modifications are present in both the terminal tails as well as in the globular core domains. Recent literature shows that the most common of these modifications are methylation, phosphorylation, acetylation, ubiquitylation and non-coding RNA. These and other modifications are maintained by various interconnected signaling pathways, involving a number of catalyzing enzymes. Some of these enzymes can catalyze the formation of specific types of PTMs (writers), while proteins that recognize and interpret certain PTMs via certain domains (readers) and enzymes that remove some of these modifications (erasers) [4,5]. Both early as well as more recent research demonstrated that these modifications can initiate and/or influence various molecular mechanisms responsible for different diseases. Over fifty years ago it was demonstrated in vitro that histone acetylation strongly suppresses transcription, inhibiting the effect of Histone incorporation into DNA [6]. This effect has been attributed to the neutralization of the positive charge of lysine residues, which alters the basi properties of histones resulting in less compact chromatin structure [7,8,9]. Consequently, histone acetylation generally correlates with transcriptional activity and the regulation of diverse cellular processes. The PTMs of histone proteins are also known to influence gene expression by altering chromatin structure, the same modifications activate a number of biological processes, including chromosome packaging, apoptosis, transcriptional activation/deactivation, and DNA damage/repair [10]. Some of these investigations have also provided strong evidence that aberrant genetic and epigenetic alterations are the hallmarks of cancer. In a relatively recent work [11], it was reported that amino acids in histone proteins located at or near key regulatory PTMs are mutated in some forms of cancer and that the protein machinery that writes, reads and erases these modifications is also frequently altered in cancers in which these PTMs can act as oncogenic drivers. These and other epigenetic findings induced enhanced research activities targeting epigenetic regulators, i.e., proteins involved in the detection and interpretation of epigenetic signals. These research efforts allowed us to decipher various molecular mechanisms, which could be translated into new therapies. It is interesting to note that therapies based on epigenetic molecular mechanisms if successful would have the advantage of acting at the transcriptional level, enabling the repression of certain genes or the transcriptional reactivation of genes epigenetically silenced in cancer. 

Resistance to chemotherapeutics, particularly in the advanced stages of some forms of cancer, remains a central point in the failure or success of a given therapy. Research efforts to understand the mechanism(s) behind such resistance are ongoing. Recent epigenetic investigations of aberrations in DNA methylation, histone modification, and non-coding RNA regulations have been extensively carried out, showing that these aberrant epigenetic regulations contribute to tumor resistance [12]. Further studies along these lines suggest that targeting these epigenetic regulators represents an opportunity for the development of a new class of therapeutics capable of limiting the influence of drug resistance on the outcome of a cure [13]. Accumulating evidence over the last ten years shows that abnormalities in the landscape of histone PTMs and DNA methylation are associated with a number of human diseases, including various forms of cancer, autoimmune disorders, certain neurodegenerative diseases and cardiovascular disorders [14,15,16]. The same studies showed that unlike genetic alterations, which are impossible to reverse, epigenetic aberrations are potentially reversible, allowing the diseased cell population to return to normal state. Such potential reversibility of epigenetic alterations has justly attracted intense research activities aimed at exploiting such characteristics to develop new therapies for various diseases. 

The investigation on histone proteins PTMS, particularly in clinical samples, was until recent years limited to the use of biochemical methods using specific antibodies. These biochemical approaches demonstrated a number of limitations, including antibody cross-reactivity arising from close similarity of certain modifications and their resulting chemical structures (e.g., mono-, di-, and trimethylation) [17]. In recent years, MS-based methods have emerged as a better alternative for an unbiased and fairly comprehensive investigation of these modifications. Most of the existing MS studies investigating histone proteins and associated PTMs have been performed using liquid chromatography (LC) coupled to mass spectrometry (MS) (LC-MS), commonly termed “shotgun” proteomics [18]. Prior to the emergence of this MS-based platform, 2D-gel electrophoresis was the method of choice for the analyses of complex protein mixtures [19]. Over the last decade, however, this powerful analytical tool has been almost displaced by the “shotgun” approach. This displacement is due to a number of limitations associated with two-two-dimensional gel electrophoresis(2DE), including large sample requirements compared to MS, relatively long analysis times, narrow dynamic range in terms of relative molecular weight (Mr)_,_ isoelectric point (PI) and lower sensitivity compared to MS. However, 2DE combined with mass spectrometry can be a powerful analytical tool for the investigation of complex protein mixtures. In our opinion an arrangement that uses 2DE as a first step of fractionation and initial identification of the components of a given protein mixture, followed by protein digestion and LC/MS-MS analyses is likely to furnish a more comprehensive picture of the composition and the identity of the individual components within the investigated mixture. The value of this setup has been demonstrated over 10 years ago [20]. The authors used 2DE in combination with nano LC and high-resolution mass spectrometry to investigate Hela cell proteins. To enhance the quantitative aspect of their study, stable isotope labeling with amino acids in cell culture (SILAC) was used. This investigation allowed the detection and identification of proteins-specific regulation and modification in the course of apoptosis. The same investigation showed that more than 50% of the spots in the 2DE map contained multiple proteins (proteoforms). This observation underlines the importance of LC coupled to MS-MS for more confident identification of proteoforms separated by 2DE. 

Some of the MS-based methods as well as 2DE are briefly discussed below. We also discuss the contribution of these analytical methods to research efforts related to drug resistance to chemotherapy and the search for potential epigenetic biomarkers. 

## 2. Discussion 

### 2.1. MS-Based Methods for the Analyses of Histone Proteins 

There are two main MS platforms, which are in use for the analyses of histone proteins and associated PTMs. The first uses liquid chromatography coupled to tandem mass spectrometry (LC/MS-MS) and is commonly referred to as “shotgun” analyses. The second and less frequently used platform uses two-dimensional gel electrophoresis (2DE) to perform preliminary, fractionation, identification and quantification of the intact proteins, followed by protein digestion and LC/MS-MS or MS or matrix-assisted laser desorption ionization-time of flight(MALDI-TOF)analyses.

### 2.2. Shotgun Analyses

This approach encompasses three different methods of analysis: “bottom-up”,” top-down” and “middle-down”. Although the three procedures are called with three different names there are minimal differences in the experimental setup (hardware). It can be said that the main difference is in the size of the peptide chains injected into the ion source of the mass spectrometer. In the case of top-down intact proteins, in bottom-up relatively short peptides (5–20 amino acids), and in middle-down a relatively long peptide chain (20–50 amino acids). In the bottom-up method, histone proteins are enzymatically digested and the resulting peptides are separated by reversed-phase LC chromatography and analyzed by electrospray ionization (ESI)-MS and MS/MS. The resulting data are used to search various protein databases, helped by highly advanced computing programs to establish the identity of the individual parent(intact) protein and associated modifications. The main steps in this method are schematically represented in Figure 1.

It can be emphasized that all three methods present a number of challenges, some of which have been partially addressed by various research groups to optimize analyses. One of these challenges is the choice of the enzyme to digest the investigated proteins. Trypsin is considered the enzyme of choice for protein digestion. However, the presence of an unusually high number of basic amino acids in the sequence of histone proteins results in an extremely high number of trypsin cleavage sites (lysine-arginine). In the bottom-up method, proteins are commonly digested with trypsin, resulting in very short peptides. This defect has two consequences: first, these short peptides cannot be retained efficiently by reversed-phase LC columns, and second, these short peptides do not retain sufficient charge states to provide meaningful fragments in the MS-MS stage. The direct result of both characteristics is the failure to capture a fair number of proteoforms in each run of analysis. To overcome both difficulties, various research groups have adapted derivatization procedures, including the use of formaldehyde [21], maleic anhydride [22] and propionic anhydride [23,24]. These derivatization methods contribute to a higher MS sensitivity and in the case of propionic anhydride derivatization, the resulting peptides are less hydrophilic.

In the top-down method, intact histone proteins are separated by reversed-phase liquid chromatography and analyzed by MS, MS/MS without prior enzymatic digestion. This approach provides more molecular specificity of the intact proteins, allowing the analysis of proteoforms (different molecular forms of a protein product of the same gene). The main limitation of the top-down method is related to the difficulty of efficient separation of the investigated proteins and the relatively poor fragmentation of their corresponding ions. That said, there are recent indications that the top-down method is developing into a relatively high throughput method, at least for proteins with Mr up to ~30 kDa. Sequence information on these proteins is obtained using collision-induced dissociation (CID) [25]. It has been also demonstrated that the use of electron capture/transfer dissociation (ECD/ETD) [26,27] could improve the sequence coverage. Furthermore, due to much lower collision energy (compared to CID), ETD seems to preserve the most common labile PTMs during fragmentation, facilitating more reliable localization of the site of modification [28]. 

In the middle-down method, relatively long peptide chains (3–5 kDa) are generated through enzymatic digestion with proteases that cleave at less frequently occurring residues, such as Glu-C and Asp-N. The resulting fragments are usually separated using weak-cation exchange (WCX) chromatography combined with hydrophilic interaction liquid chromatography. The reduced number of peptides (compared to the bottom-up method) results in less complex digests, while longer peptide chains increase the probability of detecting multiple co-occurring neighboring PTMs. One of the main limitations of this method is the ability to distinguish isobaric peptides, a problem that becomes more serious for the larger molecules analyzed in middle- and top-down methods. These isobaric species are often coeluted to the ion source and fragmented in the MS/MS phase, and as a consequence, they cannot be differentiated. Differentiating these isobaric species requires the use of differential ion mobility spectrometry. Attempts to render middle-down a high throughput method have been hindered by a number of experimental obstacles, one of which is the absence of what can be described as an ideal proteolytic enzyme that can produce controlled peptide chains in the range of 3–10 kDa. Currently, there are a number of single-residue specific proteases used in middle-down analyses, including Lys-N, [29,30] Asp-N, [31,32,33] Lys-C, [34,35,36]. These proteases are known to produce a relatively higher number of middle-range-sized peptides compared to trypsin. Attempts to optimize the yield of these peptides included chemical cleavage methods as an alternative to enzymatic proteolysis [37]. It is interesting to note that in one of these studies, the average molecular weight of the identified peptides generated by formic acid-induced digestion was similar to Asp-N and Glu-C. The same study showed that the detection of these relatively large peptides could be improved by optimizing strong-cation exchange SCX chromatography together with the use of columns packed with larger pore size materials.

#### Enhancement of MS/MS in Top/Middle-Down Methods

Over the last two decades, the role of “shot gun” in protein analyses has been enhanced by the introduction of more efficient protein/peptide fragmentation techniques as well as the addition of ion mobility to the MS platform commonly used in such analyses. Various works have demonstrated that the performance of top-down and middle-down MS-based methods for proteoform characterization can be strengthened by the use of ion mobility mass spectrometry (IM-MS). This technique separates analyte ions based on their gas-phase size and structure. Under the general title “ion mobility” there are four different configurations, including, Drift Tube Ion Mobility, Cyclic Ion Mobility, Trapped Ion Mobility spectrometry, differential Ion Mobility and Field Asymmetric Waveform Ion Mobility spectrometry. Most of these ion mobility variants operate on the same basic principle. Separation of ions in a buffer gas (normally helium or nitrogen) under the influence of an electric field. Basically, IM-MS enhances signal in tandem mass spectrometry (MS-MS). This additional component in the experimental setup for proteoform identification also enhances the detection of isomers and PTMs in the investigated sample [38,39,40,41].

One of the main difficulties in the top-down and middle-down methods is poorfragmentation in the MS-MS phase. For many years such fragmentation has been performed using high energy collision-induced dissociation (CID). In recent years a number of alternative fragmentation methods have been implemented, improving considerably the sequence coverage and resulting in more robust identification of the investigated proteoforms and associated PTMs. These alternative methods include electron transfer dissociation (ETD) [42,43], electron capture dissociation (ECD) [43], ultraviolet photodissociation [44,45] and Infrared Multiphoton Dissociation [46].

In the ECD method, low energy electrons are captured by multiply charged protein/peptidepositive ions. Such a reaction results in a relatively soft cleavage of the Cα–N bond in a peptide backbone. Such low-energy interaction allows much better localization of labile PTMs and more informative sequencing of modified peptides. This highly efficient fragmentation method has two limitations. First, efficient fragmentation requires high charge states. Second, the incompatibility of free electrons with radiofrequency-based ion trap instruments. The ECD has remained applicable exclusively in Ion Cyclotron Resonance (ICR MS). However, attempts were made to adapt this reaction to time-of-flight [47] and Orbitrap [48] instruments through the use of an electro magnetostatic cell [49]. The ETD was developed to be used in a wider range of instruments in particular RF-based ion traps [42]. In this reaction, multiply charged precursor cations are mixed with negatively charged reagent molecules to facilitate electron transfer from anions to cations resulting in the dissociation of Cα–N bonds similar to ECD.

Another technique developed alongside electron-based fragmentation is Ultraviolet photodissociation(UVPD). This method has been extensively tested in time-of-flight (TOF) instruments [50] and later in linear ion trap (LIT) instruments [51]. In UVPD, fragmentation is provoked by direct excitation of irradiated ions to their dissociative state, which results in extensive fragmentation of the amino-acid backbone while preserving most PTMs. It has been reported that a single UV laser pulse is sufficient to acquire a near 100% sequence coverage of small proteins with molecular weights below 20 kDa [52].

Current literature suggests a new trend in MS/MS analysis of proteins/peptides, where multiple regimes of induced fragmentation are available within a single instrument. A recent work [53] described an instrument configuration based on the Orbitrap Explor is 480 mass spectrometer that has been coupled to an Omnitrap platform [54]. The Omnitrap possesses three distinct ion-activation regions that can be used to perform resonant-based collision-induced dissociation, several forms of electron-associated fragmentation, and Ultraviolet photodissociation.

### 2.3. Two-Dimensional Gel Electrophoresis 2DE

The use of 2DE in combination with mass spectrometry remains a powerful analytical tool for the analysis of complex protein mixtures. in this method, the intact proteins are separated according to their molecular mass and isoelectric point. The separated proteins are displayed as individual spots on a 2D map. We have to bear in mind that a single spot on the map does not mean a single performance. The number of proteoforms within each spot cannot be established before subsequent enzymatic digestion followed by LC separation of the resulting peptides and MS-MS analyses. Analyses of these peptides can be performed using electrospray ionization or MALDI. As mentioned above. The high resolution by 2DE combined with MS-MS sensitivity and accuracy render the identification of the investigated proteoforms more comprehensive compared to exclusively MS-based methods. 

It can be argued that the experimental arrangement, schematically represented in Figure 2 requires longer analysis times and major sample consumption compared to MS-based platforms. 

Furthermore, the multiplicity of the techniques used in this platform and the difficulties in transforming such an arrangement into a fully automated system renders its transfer from the lab to the clinic fairly demanding. Despite these drawbacks, we believe that such an experimental arrangement can provide valuable information on histone proteins PTMs and DNA methylation. 

The rest of this review will focus on two arguments, drug resistance in cancer and potential cancer biomarkers associated with epigenetic modifications. The choice of both arguments is motivated by the increasing contribution of MS-based proteomics to both areas of research. 

### 2.4. Drug Resistance in Cancer 

Drug resistance remains one of the main challenges to present-day oncology. This resistance, whether intrinsic or acquired is recognized as the main obstacle to a successful cure of many forms of cancer. Advances in genomic, epigenetics, proteomics, analytical techniques and bioinformatics together with data provided by large-scale investigations of patient-derived cancer samples furnished a wealth of information on the various mechanisms of drug resistance. These research efforts made it clear that resistance to therapy is a multifaceted and complex problem. Some of the reported mechanisms believed to be responsible for drug resistance are schematically represented in Figure 3.

Attempts to combat drug resistance to therapy extended for more than 70 years and went through various phases. Early chemotherapeutics included mustard1 and aminoterin2 [55]. The early stages of this therapy resulted in fast remission of acute leukemia in children; however, this initial success was quickly followed by drug resistance and disease relapse. 

Currently, there is compelling evidence that cancer cells harbor both genetic and epigenetic alterations. In recent years large-scale investigations of patient-derived cancer samples have identified various oncogenes, which can be pharmacologically targeted to provide effective therapies for a number of cancers. These studies have shown that the initial response to these therapies is highly encouraging and some patients have even experienced complete recovery. However, the same studies have demonstrated that resistance to such therapies emerged frequently. In recent years numerous studies have linked drug resistance to certain epigenetic alterations. For example, some of these studies have demonstrated that DNA methylation can induce the suppression of a number of genes associated with cancers. For instance, retinoblastoma tumor suppressor (RB1) was identified as one of the suppressor genes that hypermethylated in tumor tissues [56]. Another study reported the silence of the DNA repair gene (BRCA1) in breast and ovarian cancers due to its hypermethylation [57]. The DNA has five methyltransferases, namely DNMT1, DNMT2, DNMT3A, DNMT3B, and DNMT3L, however, only three of these members have catalytic methyltransferase activity [58,59]. DNMT1 Is considered a maintenance methyltransferase, methylates preferentially hemi-methylated DNA and is responsible for replicating parental DNA methylation patterns to newly synthesized DNA daughter strands [60]. DNMT3A and DNMT3B are known to be more inclined to methylate unmethylated CpG dinucleotides [61].

A major step in combating drug resistance was the introduction of combination chemotherapy [62,63] to replace single-agent chemotherapy. The combined administration of agents with different mechanisms of action worked fairly well for some forms of lymphoma, breast cancer and testicular cancer [64,65]. Despite the remarkable successes of this therapy, the data accumulated over almost half a century showed that combination therapy cannot cure many forms of cancer, and other answers are needed. A better understanding of the biology of cancers and the factors and characteristics, which transform healthy cells and tissues to malignancies have underlined the need for new therapeutic strategies. A direct result of these strategies is the discovery and development of targeted therapies [66,67]. These therapies have shown high efficacy against tyrosine kinases, nuclear receptors and other molecular targets. Similar efficacy was also reported when targeting driver oncogenes that induce cancer. In the meantime, various studies have shown that this efficacy can be radically reduced in the course of treatment due to the development of acquired drug resistance. The failure of targeted therapies to overcome drug resistance for many forms of cancers underlined the need to fully understand the mechanism(s), which induce such resistance. As pointed out above, unlike genetic alterations, which are impossible to reverse, many epigenetic alterations are reversible. An example of how the reversal of dysregulated epigenetic regulators can be used to reduce drug resistance was given by Ge et al. [67] The authors used proteasome inhibitor, bortezomib as part of the treatment of multiple myeloma. The proteasome is an enzyme complex that plays an important role in the cell cycle. Proteins that are no longer needed by the cell are marked through ubiquitination and degraded by proteasome [68]. Resistance to bortezomib in the treatment of multiple myeloma is considered a major obstacle to the successful treatment of this form of cancer [69,70]. The effects of bortezomib-based intermittent therapy were compared to treatment in combination with histone deacetylase (HDAC) inhibitors on drug-tolerant multiple cells [68], and demonstrated that the combination of HDAC inhibitors and high-dosage intermittent therapies, as opposed to sustained proteasome inhibitor monotherapy, can be more effective in treating this form of cancer by preventing the emergence of proteasome inhibitor tolerant cells. This study concluded that the therapeutic basis in such treatment is the reversal of dysregulated epigenetic regulators in patients treated with proteasome inhibitors in combination with an epigenetic factor inhibitor, HDAC. 

### 2.5. MS Investigation of Proteins Implicated in Drugs Efflux

Drug efflux is considered an important mechanism responsible for multi-drug resistance (MDR) in many forms of cancer. Recent literature suggests that the most common reason for the acquisition of resistance to a broad range of anticancer drugs is the expression of one or more energy-dependent transporters that detect and eject anticancer drugs from cancerous cells. A similar role by these proteins is also documented in anti-bacterial therapies. Several cell membrane transporter proteins have been linked to resistance to commonly used chemotherapeutics as well as certain targeted therapies. These proteins are known to promote drug efflux, where the most investigated is the ATP binding cassette (ABC) transporter family. In total, there are 48 members however only three members have been extensively studied in correlation to multi-drug resistance. These are MDR1 also known as P-glycoprotein and ABCB1), MDR-associated protein 1(MRP1: also known as ABCC1) and breast cancer resistance protein (BCRP; also known as ABCG2) [71]. It is also known that every living organism has encoded within its genome many members of this family. It has been reported that when tumor cells are exposed to a cytotoxic agent, the ABC transporter expression migrates from higher expressing cells to the cells that have lower expression of the corresponding protein, which results in transforming drug-sensitive cells to resistant ones [72]. The association of higher expression of MDR proteins with chemoresistance was reported almost 30 years ago [73]. The authors reported increased MDR1 expression that correlated with poor outcomes of medulloblastoma (MB). 

In recent years, MS-based investigations provided useful information on the expression of members of this family, which are associated with MDR. Liquid chromatography coupled to tandem mass spectrometry (MS-MS) was used to assess the pattern of expression of ABC transporter proteins in the development of drug resistance in cell lines [74]. The authors used an assay called quantitative targeted absolute proteomics [75]. In their investigation, various cell lines were treated with different therapeutic agents to induce resistance. The expression of the three transporter proteins was quantified at different stages of resistance development. This investigation concluded that during the development of drug resistance, the cell line type and patch, but not the drug type were the determinant factors in the overexpression of the monitored proteins. 

In a more recent study, LC-MS-based multi-omics analyses were used to investigate the mechanism of resistance to Palbociclib [76].This therapeutic drug is a specific CDK4/6 inhibitor that has been widely used in the therapy of various forms of cancer. This inhibitor is known to target the proliferation of cancer cells through cell cycle inhibition. This inhibitor has been approved for metastatic breast cancer [77]. To investigate the acquired resistance to Palbociclib, the authors used proteomic, metabolomic, and glycoproteomic techniques to perform what they call multi-omics analyses. These analyses showed that metabolism-related pathways were generally upregulated in Palbociclib resistance cells. The same study reported an increase in the N-glycan biosynthesis in resistant cells. One of the main conclusions of this study is that the use of glycosylation inhibitors could reverse the acquired resistance to Palbociclib. 

### 2.6. Single-Cell MS to Investigate Acquired Drug Resistance

Over the last two decades, there has been an impressive increase in the development and application of single-cell genomics, transcriptomics, proteomics, and metabolomics. Single-cell metabolomics is attracting increased attention, because of its potential to provide much-needed information on the level of circulating cancer cells that lead to metastasis [78,79,80]. In recent years mass spectrometry has become a widely used method for sensitive and simultaneous detection of multiple metabolites at the single-cell level. Mass spectrometry combined with constructed machine-learning models was used to monitor chemotherapy-acquired resistance based on single-cell metabolomics [81]. The authors monitored metabolomic profiles of live cancer cells treated with low, high, and zero chemotherapy. The data generated were fed to various machine learning models to predict the degree of resistance of the individual cells. The authors reported that a systematic comparison of performance was conducted among multiple models, and the method validation was carried out experimentally. The main conclusion of this investigation is that the machine learning models constructed on the acquired MS datasets can rapidly and accurately predict different degrees of drug resistance of live single cells. 

### 2.7. Drug Resistance in Childhood Cancers

Recent landmark sequencing studies have demonstrated that genetic mutations in most childhood cancers are substantially lower than those in adult cancers [82,83]. Furthermore, fusion genes are more common than in adult cancers, and certain specific mutations found in pediatric cancers are rare in adult counterparts. Despite major improvements in clinical management including timely diagnosis, advanced supportive care and refined multimodality treatment, prognosis remains grim for various risk groups. Aberrant epigenetic regulation, i.e., changes in gene transcription not due to DNA sequence alterations, is now increasingly recognized as a fundamental process in malignant transformation, tumor progression and drug resistance. Advances in the treatment of pediatric cancers have improved survival across all tumor types. That said, drug resistance continues to limit survival for a considerable number of patients. It is interesting to note that the development of new therapeutic approaches, including tyrosine kinase inhibitors, monoclonal antibodies, and immu-noncology approaches, has resulted in yet more resistance mechanisms.

Solid tumors account for approximately 30% of all childhood cancers. Although advances in the treatment of pediatric cancers have improved survival to more than 80% across all tumor types of prognosis of many risk groups remains very poor. Multidrug resistance (MDR) represents the main challenge to the overall survival. It is reasonable to state that one of the main mechanisms behind such resistance is related to an enhanced efflux rate of chemotherapeutic drugs from tumor cells through drug transporters [83]. Medulloblastoma (MB) is the most common malignant pediatric brain tumor. Recurrence and progression of disease occur in 15–20% of standard risk and 30–40% of high-risk patients [84]. This form of cancer comprises four principal molecular groups with different patterns of metastasis and overall prognosis. The association of higher expression of MDR proteins with chemoresistance was reported almost 30 years ago [85]. The authors reported increased MDR1 expression that correlated with poor outcomes of medulloblastoma (MB). A recent work [86] investigated mechanisms associated with medulloblastoma therapy resistance, in particular the role of oncoprotein YB-1 in such resistance. Y-box binding protein 1 (YB-1), is a multi-functional protein encoded by the *YBX1* gene on chromosome 1. This oncoprotein has been implicated in almost all mRNA- and DNA-dependent processes in the cell, with recorded roles in mRNA translation and packaging, DNA repair, proliferation, pre-mRNA splicing and DNA replication [87]. The over-expression of this protein has been also reported in a number of cancers, including renal cell carcinoma [87], and breast cancer [88]. One of the main conclusions of the investigation by Taylor et al. [85] is the association between YB-1 expression and various aspects of medulloblastoma tumorigenesis, including cell invasion, MYC oncoprotein activity and lipid metabolism. The same study also found an evident correlation between high expression of the YB-1 gene and poor survival outcomes of patients in three out of four subgroups of medulloblastoma. In most forms of cancer, the initiation of metastasis is considered a form of defense by the cancerous cells to resist therapies. Metastasis of medulloblastoma occurs in one-third of patients at diagnosis and in two-thirds at the time of relapse. Furthermore, the presence of metastasis is the strongest predictor of mortality among MB patients [89]. MS-based proteomics has been used to investigate the metastasis of MB [89]. Mass spectrometry imaging (MSI) using matrix-assisted laser desorption ionization (MALDI). MSI is a label-free method that is used to map the distribution of a wide range of molecules in different biological samples. One of the key features of MALDI-MSI that make its use appealing is the ability to detect and study the distribution of multiple compounds simultaneously without the need for labelling. It is also relevant to note that the powerful and widely used LC/MS-MS lacks information on spatial distribution, which is highly important in the analyses of heterogeneous biological samples such as cancerous tissues. 

MALDI-MSI was used to investigate whole brains from a mouse model of human medulloblastoma [90]. The analyses were performed within the tumor area and in the surrounding normal areas. The authors reported the detection of spatially heterogeneous lipids within the cancerous region. Furthermore, boundaries of metastasizing and non-metastasizing primary tumors were readily defined allowing the identification of lipids associated with the MB metastasis, including phosphatidic acids, phosphatidylserines, phosphoinositides and phosphatidylethanolamines. 

Acute lymphoblastic leukemia (ALL) accounts for ~30% of all cancers in children, making it the most common childhood cancer [91,92]. Despite the successful treatment rate of about 70–80%, the main challenge remains the risk of relapse for about 15–20% of the patients [93,94]. Existing evidence indicates that the cure rate is strongly related to an early detection of the disease and a correct stratification of the various risk groups. The lack of clear initial symptoms renders a confident clinical diagnosis of the disease rather challenging. Currently, such diagnosis requires a bone marrow biopsy/aspiration, which is highly invasive and extremely distressing in particular for childhood patients. 

In a well-designed investigation [95], the authors performed multi-omics profiling of childhood ALL cell lines. A total of fifty-one cell lines (29 B-linage and 22 T-linage). Profiling of these cell lines as well as response to drugs were performed using proteomics, transcriptomics, and pharmacoproteomic techniques. Liquid chromatographs coupled to tandem mass spectrometry together with peptide labeling using tandem mass tags (TMT) [95] were applied to quantify both peptides and proteins. These analyses allowed the quantification of more than 12,000 proteins and 19,000 protein-coding transcripts. Drug sensitivity of the investigated cell lines was also examined for more than 500 oncology drugs. The authors reported that their proteomics-guided analyses of 49 childhood ALL cell lines, generated a comprehensive resource of biomarkers and drug sensitivities as well as the identification of potential therapeutic vulnerability to target the MEF2D/HNRNPUL1 fusion protein in a high-risk subtype of the disease. 

The role of cancer stem cells (CSCs) as one of the mechanisms of resistance in pediatric cancers continues to attract intense academic as well as clinical research to decipher this mechanism, which may lead to a new generation of therapeutics. These stem cells are known to have the capacity to initiate a new tumor, a slower proliferation rate and/or maintenance of a basal undifferentiated population. These features can inherently induce therapy resistance, as most chemotherapeutic agents rely on damage to the mechanisms of cell proliferation. There is some evidence that markers of these cells can initiate therapeutic resistance. A representative example of such a situation is CD133, a surface marker associated with stem cells in a number of pediatric solid tumors and leukemias [96] can promote therapeutic resistance and has been associated with poor survival [97]. Present-day literature suggests additional multidisciplinary research is needed to shed further light on the role of CSCs in drug resistance and the aggressiveness of the disease [98]. 

## 3. Potential Epigenetic Biomarkers Based on MS Analyses

For a number of years, the utility of biomarkers such as proteins, lipids and metabolites has been mainly confined to the area of disease diagnoses. More recent advances in genetic, epigenetic and proteomics research identified other areas in which the same type of biomarkers can play relevant roles in diagnosis, prognosis, and patient stratification. Furthermore, many marker proteins identified under various disease conditions have become privileged targets of various pharmaceutical companies in the search for new therapies. 

Epigenetic biomarkers associated with DNA methylation, histone modifications, and noncoding RNAs, are considered driver factors in a number of diseases, including various forms of cancer. It has been reported in numerous publications that aberrant DNA methylation can lead to silencing of crucial tumor suppressor genes or upregulation of oncogene expression. Furthermore, the dysregulation of histone post-translational modifications (PTMs) and chromatin spatial organization, which affect transcription, regulation of gene expression and DNA repair have been associated with several cancers. Epigenetic biomarkers provide valuable information on some mechanisms of drug resistance, such information is also considered relevant to research efforts to discover and develop new therapies for various diseases. For many years the detection and quantification of histone proteins circulating in plasma was limited to enzyme-linked immunosorbent assay (ELISA) for the detection of antihistone antibodies [99,100]. However, the poor reproducibility between different platforms and the low sensitivity of this approach paved the way for the use of MS-based analyses [101] 

The MS-based strategy was developed to detect circulating histones H3 and H2B in plasma [102]. The authors used tandem mass spectrometry in multiple reaction monitoring (MRM) mode to examine plasma samples derived from under 20 patients with confirmed bacteriaemia septic shock and 10 healthy controls. Quantification of the detected histones H3 and H2B showed significantly higher levels in the patient’s plasma. Based on these results, the authors suggested that their method of analysis could be used for early septic shock diagnoses and the prognosis of fatal outcomes. Considering the conclusions of this investigation the following observations can be made: First, these results are a good example of proof of concept. However, before considering these data as potential biomarkers, a much higher number of samples is needed to validate these results, which are based on a very limited number of subjects. Second, histones, as well as their covalent modifications are known to circulate in blood samples derived from cancer patients. In their study, the authors do not specify whether the detected histones were modified or not. Third, high levels of mono- and oligonucleosomes are commonly detected in the blood of patients with malignant tumors [103]. Research efforts to establish a significant correlation between high levels and the severity of the disease are not giving the desired results. Data acquired so far indicate that high levels of circulating nucleosomes can be induced by either benign or malignant without reasonable differentiation. A comparison of the levels of circulating histones in cancer patients to those obtained in patients with numerous benign diseases has shown that the difference is not statistically significant [104].

Increasing levels of circulating nucleosomes/histones have been detected in the blood samples from patients suffering from breast, colorectal, and prostate cancers [105,106]. Monitoring the level of circulating nucleosomes can predict tumor responses to chemotherapeutic agents in various cancer types [107]. Cell Death Detection (plus)-ELISA was used [103] to quantify circulating mono and oligonucleosomes in blood samples derived from over 400 patients with malignant tumors, over 100 patients with benign disease, and 63 healthy subjects. The authors reported that compared to the healthy subjects, the circulating levels of nucleosomes were almost 9 times and 7 times higher in malignant and benign patients, respectively. The same study concluded that the concentration of nucleosomes in serum can be a useful tool for monitoring the biochemical response during antitumor therapy, especially efficacy. In a more recent study [107], the authors used MS-based proteomics to profile circulating nucleosomes in plasma. Intact circulating nucleosomes were captured using immunoprecipitation. These captured nucleosomes were subsequently analyzed by LC coupled to MS-MS. Samples were obtained from 9 patients with colorectal cancer (CRC) and 9 healthy controls. The authors identified a number of PTMs, including methylation of histone H3K9 and H3K27, acetylation of histone H3 and citrullination of histone H2A1R3 were upregulated in plasma of CRC patients. Considering the experimental procedure and the conclusions of this study, the following observations can be made. This is a well-designed strategy to favor the detection of low-copy proteoforms in samples known to be dominated by few proteins, including albumin. The low number of samples used in this study confirms the difficulty of obtaining clinical samples from both patients and controls. However, to describe some of the observed PTMs as potential biomarkers for CRC requires the investigation of many more samples followed by validation of such markers at the clinical level.

## 4. Conclusions 

Data generated so far on the use of MS-based proteomics demonstrate notable achievements of this platform in terms of the characterization of histone PTMs both known and unknown. However, the same reports indicate that the analytical methods used in these analyses frequently fail to capture a fair percentage of the targeted proteoforms within the investigated digests. This defect becomes more evident in the analyses of histone tails, because of their hydrophilic nature and the more frequent PTMs resulting in an extremely high number of possible proteoforms. We believe that future improvements in the separation and labelling protocols together with more powerful MS detectors accompanied with enhanced mass accuracy will result in more comprehensive solutions in the analyses of histone PTMs provided by MS-based proteomics.

Some of the works discussed in this review indicate that childhood cancers differ from adult cancers by having fewer genomic mutations, on the other hand, epigenetic dysregulations such as DNA methylation and histone PTMs tend to have a greater impact on the biology of diseases in childhood. These observations suggest that the present and future search for new therapies in the area of childhood cancers is likely to be more biased towards a deeper understanding of the mechanisms associated with epigenetic dysregulation.

It is now well-established that cancer cells harbor both genetic and epigenetic alterations. This implies that current therapeutic approaches as well as those in the near future will have to consider both genetic and epigenetic changes that can be targeted and/or induced in response to therapy. The multiplicity of mechanisms associated with drug resistance, heterogeneity of cancer and numerous genetic and epigenetic alterations associated with cancer render the search for more specific and effective therapies with negligible side effects extremely challenging. To approach such an objective, we need to gain deeper insight into tumor biology, a much better understanding of the mechanisms behind drug resistance and a more comprehensive evaluation of both genetic and epigenetic alterations. Such a daunting task requires close collaboration between scientists and clinicians around the world. 

## Figures and Tables

**Figure 1 medicina-59-01722-f001:**
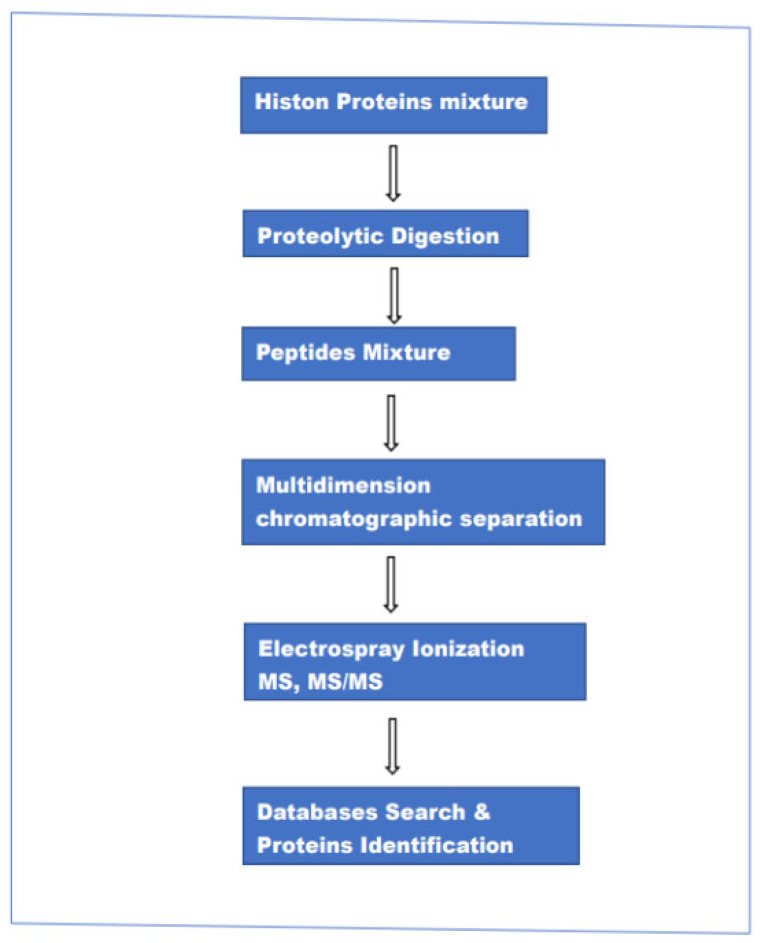
The main steps in the bottom-up MS-based method for the analysis of proteins mixture.

**Figure 2 medicina-59-01722-f002:**
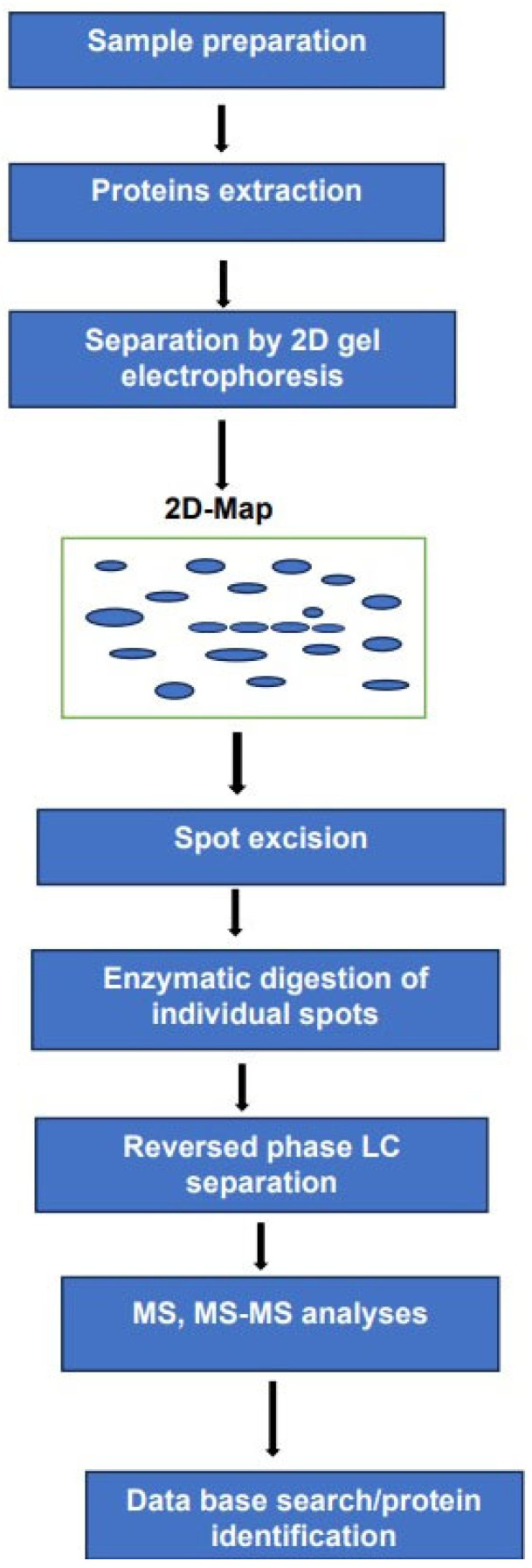
Two-D gel electrophoresis combined with LC-MS, MS/MS for the analyses of complex mixture of proteoforms.

**Figure 3 medicina-59-01722-f003:**
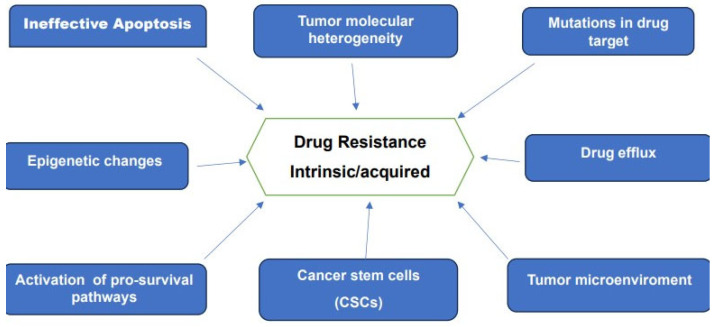
Various mechanisms associated with drug-resistance to chemotherapy and targeted Therapies.

## Data Availability

Not applicable.

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
