# Peer review of "Mass Spectrometry-Based Proteomics: Analyses Related to Drug-Resistance and Disease Biomarkers"

_medicina, 2023, doi:10.3390/medicina59101722_

Round 1

Reviewer 1 Report

Manuscript describes the use of MS based techbiques in proteomics research, which is a very broad topic. The topic is intresting and could potentially make a good impact on the community. However, the manuscript is not well organized and lacks depth of discussion. While the main topic is the use of MS based techniques, this is discussed in rather a shallow way and leaves a lot for improvement. Proper organization is needed and a much more in depth coverage of the various MS techniques used in the field should be covered. In it's current form, the manuscript does not qualify for publication.

No concerns regarding language.

Author Response

We wish to thank the referee for his valid suggestions, following which we have modified the manuscript. The last mass spectrometric approaches have been described in a new section "Enhancement of MS/MS in top/middle down methods" with the related references (18 new references on the newest MS approaches).

Reviewer 2 Report

General comment:

This review manuscript covers recent work on both drug resistance and cancer biomarkers through the use of LC-MS/MS proteomic targeting mostly nuclei-based histone literature. The authors cover very well the literature in both fields and explain very well each step of the way the most recent findings. While the drug resistance and cancer biomarkers part is complete I feel some small addition to the mass spectrometry side is needed (see comments bellow).

Comments:

1.       Page 4 line 162-166

When talking about top-down proteomics, the authors cover some of down side of CID and talk about the use of ECD/ETD. One recent development in top-down proteomics has been the use of ultra-violet photo-dissociation (UVPD) and the authors do not mention this new technology. A couple of sentences with 1-3 citations could make for a good addition to the manuscript since UVPD brings additional coverage with lower charge state proteins.

2.       Page 4 line 177-181

The authors cover briefly ion mobility spectrometry for middle-down proteomics but doesn’t cover its use in bottom-up where faster gradient on the LC-MS side shortens analytical time without affecting the number of detected peptides. Again a few sentences on this with proper citations would greatly improve the quality of this manuscript.

Author Response

The valid suggestions given by the referee have been followed in the revised version. In particular the use of UVPD has been inserted in a new section and the related references have been cited. 

Reviewer 3 Report

1. The general outline of the main text topics is poorly organized in logic. The title, keywords, introduction, and headings are very confusing in the review contents. The title emphasized the use of MS in drug-resistance and disease biomarkers (Genes, metabolites and lipids also count). It says nothing about proteomics. But the introduction is only focused on histone, PTM, and proteomics. The sections about the MS technical advancements and their applications are twisted. The introduction about the MS technique is too brief and the introduction about their biological/proteomic applications involved too many irrelevant fields which lack logical connection (childhood cancers, PTM, single-cell, drug resistance). It is suggested to only focus on one central topic.

2. The section of “2.1. MS-based methods for the Analyses of Histone Proteins” discussion is too short to provide any useful information about the MS-based method. Because the heading is about the MS-based method, authors should provide more information about how many types of mass analyzers have been conventionally used in proteomics such as TOF, orbitrap, FT-ICR, etc. The advantages and disadvantages of these mass analyzers when being used in which situation, or study purpose. The ESI and MALDI, as classical ambient ionization and in situ ionization techniques which win the Nobel prize for their breakthroughs in the protein analysis, deserves the special acknowledgments. The technical features of ESI and MALDI in generating different types of protein/peptide ions were also needed to be mentioned.

3. In terms of the LC part, there is an interchangeable use of LC and nano-LC. But in actual practice, these two systems have differences in the column/capillary size, separation performance, flow rate, and back pressure. Even for the LC method, it can also be divided into the high performance (HPLC) and ultra-high performance (UPLC) based on their particle sizes. Authors should elaborate each technique’s features and advantages in the proteomics application.

4. Typo or term misuses

Line 69, “cancer. remains…” should be “cancer, remains…”

Line 72, “showing. that these” should be “showing that these”

Line 92-93, “using mass spectrometry (MS) coupled to liquid chromatography (LC),” should be “using liquid chromatography (LC) coupled to mass spectrometry (MS) (LC-MS)”.

Line 94, “Prior to the emergence of this. MS-based platform,” should be “Prior to the emergence of this MS-based platform,”

Iine 97, please provide the full name of 2DE (in the line 94, 2D- gel electrophoresis??) before direct using the abbreviation.

Line 99, “molecular mass (Mr)” contradicts with the definition given in line 544, “Mr, relative molecular weight”.

Line 103, “LC/MS-MS” should be “LC-MS/MS”, Line 113, “MS-MS” should be “MS/MS”, so are these terms throughout the main text.

Line 124, “MALDI-MS”, to be more accurately, should be “MALDI-TOF”.

Line 131, please define the “AA” (amino acid?) before its usage.

Line 133, “electrospray MS and MS-MS”à “electrospray ionization (ESI)-MS and MS/MS”

Line 144, “very. short peptides” à “very short peptides”

Line 322, “MS of Single-cell”---"Single-cell MS”.

Line 382, “matrix assisted laser desorption Ionization (MALDI)”, full name of MALDI should show up at the first place somewhere around line 124.

There are many typos, and random or informal uses of professional terms, which need rigorously proofed. 

Author Response

The original version of the review has been modified following the referee suggestions. A new section has been inserted (Enhancement of MS/MS in top/middle-down methods, with the related-18- references) and the typo errors have been amended.

Round 2

Reviewer 3 Report

The review appreciates the authors' efforts in improving the manuscript.

A minor proofing might be needed.

Author Response

All the requirements have been satisfied.